# Risk and Prognosis of Thyroid Cancer in Patients with Graves’ Disease: An Umbrella Review

**DOI:** 10.3390/cancers15102724

**Published:** 2023-05-11

**Authors:** Marco Palella, Francesca Maria Giustolisi, Adriana Modica Fiascaro, Martina Fichera, Antonella Palmieri, Rossella Cannarella, Aldo E. Calogero, Margherita Ferrante, Maria Fiore

**Affiliations:** 1Department of Medical, Medical Specialization School in Hygiene and Preventive Medicine, Surgical Sciences and Advanced Technologies “G.F. Ingrassia”, University of Catania, Via Santa Sofia 87, 95123 Catania, Italy; markopa92@hotmail.it (M.P.); francesca_gi@libero.it (F.M.G.); adri.modfias33@gmail.com (A.M.F.); ficheramartina28@gmail.com (M.F.); antonellapalmeri.ap@gmail.com (A.P.); 2Department of Clinical and Experimental Medicine, University of Catania, 95123 Catania, Italy; rossella.cannarella@phd.unict.it (R.C.); aldo.calogero@unict.it (A.E.C.); 3Glickman Urological & Kidney Institute, Cleveland Clinic Foundation, Cleveland, OH 44195, USA; 4Department of Medical, Surgical and Advanced Technologies “G.F. Ingrassia”, University of Catania, Via Santa Sofia 87, 95123 Catania, Italy; marfer@unict.it

**Keywords:** Graves’ disease, thyroid cancer risk, thyroid cancer prognosis, umbrella review

## Abstract

**Simple Summary:**

Graves’ disease (GD) is the most common cause of thyrotoxicosis due to autoimmune hyperthyroidism, especially in women. This umbrella review aimed to provide an evidence-based summary of epidemiological studies conducted on the association between GD and the risk of developing thyroid cancer risk and its prognosis. Strong evidence was found for thyroid cancer risk in GD patients and nodular thyroid disease and mortality risk from thyroid cancer in GD patients, particularly in Europe. However, the results of this umbrella review should be taken with caution; as the evidence comes mainly from retrospective studies, the potential concerns are selection and recall bias.

**Abstract:**

Graves’ disease (GD) is an autoimmune disease considered the most common cause of hyperthyroidism. Some studies have investigated its relationship with the risk and prognosis of developing thyroid cancer. Considering that there is no consensus on the relationship between GD and thyroid cancer risk, this umbrella review aimed to summarize the epidemiologic evidence and evaluate its strength and validity on the associations of GD with thyroid cancer risk and its prognosis. This umbrella review was performed using the Preferred Reporting Items for Systematic Reviews and Meta-Analyses (PRISMA) guidelines. We systematically searched PubMed and Scopus from January 2012 to December 2022. The strength of the epidemiological evidence was graded as high, moderate, or weak by the Measurement Tool to Assess Systematic Reviews (AMSTAR-2). “Strong” evidence was found for the risk of thyroid cancer in GD patients with thyroid nodular disease (OR: 5.30; 95% CI 2.43–12) and for the risk of mortality from thyroid cancer in these patients (OR 2.93, 95% CI 1.17–7.37, *p* = 0.02), particularly in Europe (OR 4.89; 95% CI 1.52–16). The results of this umbrella review should be interpreted with caution; as the evidence comes mostly from retrospective studies, potential concerns are selection and recall bias, and whether the empirically observed association reflects a causal relationship remains an open question.

## 1. Introduction

Graves’ disease (GD) (Flajani–Basedow–Graves’ disease) is the most common cause of thyrotoxicosis due to autoimmune hyperthyroidism [1]. It is 5–10 times more common among women between the ages of 30 and 60 [2]. The prevalence of GD is approximately 0.5% in the general population, with a lifetime risk of 3%for women and 0.5% for men [3].

Thyroid cancer is the most common malignancy of the endocrine system and represents the eighth most diagnosed cancer worldwide [4,5]. Most of these arise from the follicular epithelium and include papillary, follicular, and anaplastic cancers [6]. Only a small percentage of thyroid cancers are represented by anaplastic carcinoma that arises from the parafollicular cells [7]. Several risk factors are associated with thyroid cancer development, such as exposure to primary and secondary ionizing radiations in childhood or adolescence and iodine deficiency [4].

Thyroid autoimmunity and thyroid cancer may coexist, although a pathogenic relationship has not been clearly established. However, the association between inflammation and carcinogenesis has already been recognized [4]. Furthermore, the expression of a biomarker of oxidative DNA damage, which may promote the development of papillary thyroid carcinoma (PTC), was found in both Hashimoto’s and Graves’ disease patients [5]. Although an association between thyroid cancer and Hashimoto’s disease was found, there are no definitive studies investigating the association of this cancer with GD [8]. In patients with GD, the prevalence of thyroid cancer appears to be higher, although it varies widely. Yoon and colleagues found a 2.5-fold higher risk of differentiated thyroid cancer (DTC) in GD patients than in the general population [9]. Initial evidence suggested that GD had a protective effect against thyroid cancers [10,11], but more recent studies have shown a higher prevalence of thyroid cancer, particularly in surgically treated patients [12,13]. Nowadays, many studies focusing on the clinical–pathological findings and prognosis of DTC in patients with GD have reported inconsistent or unclear data [14,15]. Additional evidence has reported higher aggressiveness and a higher risk of thyroid cancer recurrence in GD patients [16]; conversely, others have shown no differences in clinical characteristics or outcomes [17,18], whereas another study found a better prognosis and longer disease-free survival [19]. Because of these controversial findings, we conducted an umbrella review to evaluate the quality and validity of the evidence for the association between GD and thyroid cancer risk and its prognosis (mortality and recurrence/persistence).

## 2. Materials and Methods

### 2.1. Umbrella Review Methods

The umbrella review enhances the level of currently available scientific evidence by using an explicit and systematic method to compare and synthesize the findings of previously published systematic reviews/meta-analyses [20].

This umbrella review is registered with PROSPERO (CRD42023404371) and was conducted following the guidelines of the Preferred Reporting Items of Systematic Reviews and Analyses (PRISMA) [21,22], in line with the a priori protocol agreed upon by all authors.

### 2.2. Search Strategy and Selection Criteria

PubMed, Web of Science, and Scopus were searched in November 2022 to retrieve all systematic reviews and meta-analyses focusing on the risk and prognosis of thyroid cancer in patients with GD. Keywords were “Graves’ disease”, “thyroid cancer”, “risk of thyroid cancer”, and “thyroid neoplasm”. All these words were matched together using Boolean operators to create the final search string. All references in identified studies were manually searched to identify additional eligible articles. Four investigators (F.G., M.P., A.P., and M.Pu.) independently screened titles and abstracts and checked full texts. All disagreements were resolved by a fifth author (M.F.) through discussion.

### 2.3. Eligibility Criteria

Eligible studies were systematic reviews including meta-analyses of human observational studies published from 2012 to 2022. We included studies evaluating the association between GD and thyroid cancer and its prognosis (mortality and recurrence/persistence) reporting effect size (both raw and adjusted ORs). No restrictions were applied to the age, sex, type of thyroid cancer, or treatment of the participants.

Exclusion criteria were (1) non-English articles, (2) articles whose full text was not available, (3) articles in which thyroid cancer risk in patients with GD was not the primary outcome [5,23,24,25], (4) articles discussing the association between thyroid cancer and radioiodine-treated disease [26,27], (5) articles in which there was no evidence of clear statistical methodology used [28,29,30], (6) articles including case reports [31,32] and/or rare histological types of thyroid carcinoma [33].

### 2.4. Data Extraction

Four authors (F.G., M.P., A.P., and M.Fic.) independently performed the extraction, and a fifth author (M.F.) resolved the discrepancies by discussion. The name of the first author, year of publication, number and design of the study in meta-analyses, number of cases with GD and of thyroid cancer, gender by comparison group, comparison groups, age, DTC type, country, period of primary study publication, exposure time period/follow up, effects size (OR and 95% confidence interval), heterogeneity, and publication bias were extracted from each eligible article.

### 2.5. Evaluation of the Strength of Evidence

Strength of association, heterogeneity, meta-analysis *p*-value, and publication bias were used to evaluate the strength of evidence [34]. Evidence was classified in different ways, as described in Table 1.

### 2.6. Quality Assessment

Two researchers (M.P. and M.F.) assessed study quality using the Measurement Tool to Assess Systematic Reviews (AMSTAR-2), which consists of 16 items that measure the methodological quality of systematic reviews [37]. The complete description of AMSTAR-2 is reported in the article by Shea and colleagues [37]. Of the AMSTAR-2 items, seven were considered critical to quality and were known as “critical domains”. These seven “critical domains” are modifiable by the authors according to their point of view relative to the articles that have been searched [22].Therefore, we replaced critical domain number 7 (Did the review authors provide a list of excluded studies and justify the exclusions?) with number 8 (Did the review authors describe the included studies in adequate detail?*), highlighted by an asterisk, considering critical this more appropriate to assess the quality of the studies (Table 2). Based on this, we used AMSTAR-2 to define the quality of the meta-analyses included in this umbrella review. In summary, we considered studies with more than one critical flaw with or without non-critical limitations to be “critically low”, those with a critical flaw with or without non-critical weakness to be “low”, those with more than one non-critical classification to be “moderate”, or those with no or non-critical weakness to be “high” [37].

### 2.7. Statistical Analysis

The estimation of the summary effect (odds ratio) and the 95% confidence interval were calculated using fixed and/or random effects methods as appropriate. The heterogeneity between studies was assessed with Cochran’s Q test and the I^2^ statistic, which describes the percentage of the variability in effect estimates that is due to heterogeneity rather than chance. A rough guide to interpretation is as follows. I^2^ test interpretation: from 0 to 40%, might not be important; from 30 to 60%, may be moderate; from 50 to 90%, may be substantial; from 75 to 100%, may be considerable. In general, the test is conservative and so a non-significant result cannot be interpreted as showing that there is no heterogeneity. For this reason, the cutoff *p* < 0.10 was used rather than *p* < 0.05 to indicate heterogeneity [43]. The assessment of small study effects, which is an indication of publication bias, was examined by Egger’s regression asymmetry test. Finally, sensitivity analyses were conducted using the summary fixed and random effects estimates as alternative plausible effect sizes.

## 3. Results

### 3.1. Search Strategy Outcome

The initial search retrieved 494 studies. A total of 328 studies were screened based on titles and abstracts, and 307 were excluded; thus, 21 full-text studies were considered potentially eligible for inclusion. Sixteen studies were excluded due to the absence of statistical data (*n* = 3), case reports (*n* = 1), and no inherent topics (*n* = 12). Therefore, finally, five studies were included in this review [38,39,40,41,42]. The entire process of article collection, screening, and eligibility assessment is shown in Figure 1.

### 3.2. Quality Assessment and Bias

We used the 16-item AMSTAR-2 tool to assess the methodological quality of the five included meta-analyses of observational studies (Table 2). Overall, the studies scored from 10 to 12 positive/yes answers, 3 studies were rated as moderate quality, 2 studies were rated as low quality, and no studies were rated as high quality (Table 2). Publication bias was present in two meta-analyses [38,42], one meta-analysis had no publication bias [40], and two meta-analyses did not report it [39,41].

### 3.3. Risk and Prognosis (Mortality and Recurrence/Persistence) of Thyroid Cancer in Patients with Graves’ Disease

The results regarding the risk and prognosis (mortality and recurrence/persistence) of thyroid cancer in patients with GD are presented below, one after another, and summarized in Table 3 by comparison groups.

#### 3.3.1. Risk of Thyroid Cancer in Patients with Graves’ Disease by Comparison Groups

##### Risk of Thyroid Cancer in Patients with Graves’ Disease vs. Multinodular Toxic Goiter (MTG), Uninodular Toxic Goiter (UTG), or Unspecified Toxic Nodular Goiter (uTNG)

There was “modest” evidence of thyroid cancer risk in GD patients compared to MTG patients, and we found no evidence among all the others.

In particular, Staniforth and colleagues [40] found no significant risk of DTC in patients with GD vs. any type of toxic nodular goiter (OR 0.89; 95% CI 0.63–1.26, I^2^ 28.6%, *p* = 0.10) or UTG (OR 0.96; 95% CI 0.58–1.57; I^2^ 5.13%, *p* = 0.39). Conversely, a risk of DTC in patients with GD compared to patients with MTG was found (OR 1.24; 95% CI 0.81–1.90; I^2^ 0.0%, *p* = 0.82).

Since the last comparison has a risk associated with the lower limit of the confidence interval greater than 70, caution should be exercised. Finally, the risk of developing DTC in patients with GD was compared with those having uTNG, and no significant differences were found (OR 0.43, 95% CI 0.14–1.33), with substantial heterogeneity (I^2^ 71.7%, *p* = 0.01).

Jia and colleagues [42] assessed the risk of developing incidental thyroid cancer in patients with GD vs. those without GD and reported no significant difference (OR: 1.0; 95% CI 0.68–1.46) without significant heterogeneity (I^2^ 12%, *p* = 0.33). Furthermore, the same risk was evaluated in patients with GD compared to patients with toxic adenoma or thyroid nodular goiter (UTG plus MTG); these cases also showed neither a significant difference (OR 0.53 and CI 95% 0.21–1.36; OR 1.01 and CI 95% 0.65–1.57) nor significant heterogeneity (I^2^ 40%, *p* = 0.17; I^2^ 5% and *p* = 0.39). Finally, they evaluated the risk of developing PTC in patients with GD compared to those without GD and found no significant difference (OR 0.79 and CI 95% 0.24–2.64, I^2^ 0%, *p* = 0.97).

##### Risk of Thyroid Cancer in Patients with Graves’ Disease with Nodular vs. Those without Nodules

There was “strong” evidence of thyroid cancer risk in patients with GD and thyroid nodular disease compared to those with GD but without thyroid nodules (OR: 5.30; 95% CI 2.43–12) [39]. The same result was confirmed when the comparison was performed excluding patients with suspicious or malignant cytology (OR 4.02 and 95% CI 1.24–13). In both cases, there was considerably significant heterogeneity (I^2^ 83%, *p* = 0.00; I^2^ 89%, *p* = 0.00).

##### Number of Thyroid Nodules in Graves’ Disease Patients and Risk of Differentiated Thyroid Cancer

There was “modest” evidence of thyroid cancer risk in GD patients with solitary nodules compared with those who have multiple nodules. In particular, Papanastasiou and colleagues [39] found no significant risk of DTC in patients with GD and solitary nodules compared to patients with GD and multiple nodules (OR 1.39 and 95% CI 0.85–2.29, I^2^ 0%, *p* = 0.76). Since the comparison has a risk associated with the lower limit of the confidence interval greater than 70, caution should be exercised.

#### 3.3.2. Prognosis (Mortality) of Thyroid Cancer in Patients with Graves’ Disease by Comparison Groups

##### Graves’ Disease vs. No Graves’ Disease Hyperthyroid Patients

There was “no evidence” of different thyroid cancer mortality risks between hyperthyroid patients with or without GD. Two meta-analyses [38,41] compared the risk of mortality from thyroid cancer between hyperthyroid patients with GD or without GD, reporting no significant difference (OR 0.79 and 95% CI 0.17–3.67, I^2^ 0%, *p* = 0.50; OR 1.36 and 95% CI 0.19–9.82, I^2^ 0%, *p* = 0.43).

##### Graves’ Disease Hyperthyroid Patients Compared with Euthyroid Subjects

There was “moderate” evidence of thyroid cancer mortality risk between hyperthyroid GD patients and euthyroid patients. In particular, Mekraksakit and colleagues [38] compared the mortality risk from thyroid cancer between patients with GD and euthyroid patients and found a non-significant difference (OR 2.69 and 95% CI 0.70–10) associated with low and non-significant heterogeneity (I^2^ 22.9%, *p* = 0.27). Conversely, Song and colleagues reported a significant risk (OR 3.99 and 95% CI 1.19–13.4), but with significantly high heterogeneity (I^2^ 78%, *p* = 0.03) [41].

##### Graves’ Disease Patients Compared with Those without Graves’ Disease (Including Both Euthyroid and Hyperthyroid Patients)

There was “strong” evidence of thyroid cancer mortality risk between patients with and without GD in all included studies, whereas a “modest” risk was found in Asia. In particular, Song and colleagues [41] compared the risk of mortality from thyroid cancer in patients with and without GD and found a significant difference (OR 2.93, 95% CI 1.17–7.37, *p* = 0.02), with moderate, non-significant heterogeneity (I^2^ 33%, *p* = 0.20). This difference was more evident when studies reporting low incidental cancer rates were excluded (OR 7.17, 95% CI 2.14–24; I^2^ 0%, *p* = 0.51). Furthermore, stratifying by continent, Song and colleagues [41] found that significant risk persisted only in Europe and not in Asia (OR 4.89; 95% CI 1.52–16, I^2^ 38%, *p* = 0.20; OR 1.13, 95% CI 0.21–6.13, I^2^ 0%, *p* = 0.34).

#### 3.3.3. Prognosis (Recurrence/Persistence) of Thyroid Cancer in Patients with Graves’ Disease by Comparison Groups

##### Graves’ Disease Compared to Non-Graves’ Disease Hyperthyroid Patients

There was “moderate” evidence of a higher risk of recurrence/persistence/worse prognosis in hyperthyroid patients with or without GD. In particular, two meta-analyses compared recurrence/persistence/worse prognosis between hyperthyroid GD patients and patients without GD, reporting an increased OR in both studies (OR 2.66, 95% CI 0.94–7.54; OR 3.56, 95% CI 1.18–11), even if with non-significant heterogeneity (I^2^ 1.8%, *p* = 0.41; I^2^ 5%, *p* = 0.30) [38,41]. Since the first comparison has a risk associated with the lower limit of the confidence interval greater than 70, caution should be exercised.

##### Patients with Graves’ Disease Compared with Euthyroid Patients

There was “no” evidence of a higher risk of recurrence/persistence/worse prognosis between GD patients and euthyroid patients. In particular, two meta-analyses compared recurrence/persistence/worse prognosis between patients with GD and euthyroid patients, and both did not show significant differences (OR 1.39, 95% CI 0.52–3.76; OR 0.86, 95% CI 0.42–1.77), with substantial and considerably significant heterogeneity, respectively (I^2^ 64.2% and *p* = 0.04, I^2^ 93%, *p*= 0.00) [38,41].

##### Patients with Graves’ Disease Compared to Non-Graves’ Disease Patients (Including Both Euthyroid and Hyperthyroid Patients)

There was “weak” evidence of a higher risk of recurrence/persistence/worse prognosis between GD vs. non-GD patients which became “moderate” after stratification by continent [41]. In particular, Song and colleagues evaluated the recurrence/disease progress of DTC between patients with GD and patients without GD and found a non-significant difference (OR 1.07, 95% CI 0.51–2.22) with substantial significant heterogeneity (I^2^ 65%, *p* = 0.00). Furthermore, they reported a non-significant result when stratifying for medium-quality vs. high-quality studies (OR 1.50, 95% CI 0.60–3.79, I^2^ 64%, *p* = 0.07). Similarly, stratifying by continents, they reported a significant “moderate” risk in Europe and America (OR 1.74, 95% CI 1.02–2.98), while in Asia, it would appear not to be a risk factor (OR 0.43, 95% CI 0.25–0.77). Moreover, by eliminating studies on incidental cancer, they reported a “moderate” significant difference between the two groups (OR 1.75; 95% CI 1.04–2.95), without heterogeneity (I^2^ 0%, *p* = 0.81) [41].

## 4. Discussion

### 4.1. Main Findings and Interpretation Considering Evidence

This umbrella review, reporting data from five meta-analyses [38,39,40,41,42] including both retrospective and prospective studies, suggests that there is:“Strong” evidence of thyroid cancer risk in patients with GD and thyroid nodules compared to patients with GD without nodules.“Modest” evidence of thyroid cancer risk in GD patients compared to MTG patients and in GD patients with solitary nodules compared with GD patients with multiple nodules.“Strong evidence of increased thyroid cancer risk mortality in GD patients compared with non-GD patients (including both euthyroid and hyperthyroid patients) that increased after excluding the low-rate incidental cancers and also after stratifying by continent (Europe higher than Asia) [41].“Moderate” evidence of thyroid cancer mortality risk among patients with GD compared with euthyroid patients [38,41].“Moderate” evidence of higher risk of recurrence/persistence among patients with GD compared to those without GD (including both euthyroid and hyperthyroid patients) by continents (Europa and America vs. Asia).“Moderate” evidence of a higher risk of recurrence/persistence in hyperthyroid patients with GD than in those without GD [38,41].“No” evidence of different thyroid cancer mortality risks between GD and non-GD hyperthyroid patients [38,41].“No” evidence of a higher risk of recurrence/persistence between patients with GD compared with euthyroid patients [38,41].

Thyroid nodules are a frequent clinical finding, ranging in infrequency from 19 to 69%, in subjects undergoing random control by ultrasound scan, and are more frequent in women and the elderly. The importance of a proper diagnosis and follow-up of thyroid nodules is because in 7–15% of cases, they are cancers [44]. A review of the literature reported that patients with GD may have a higher prevalence of both benign nodules and DTCs [45]. In our umbrella review, we found “modest” evidence for the risk of developing DTC among patients with GD and solitary nodules compared with GD patients with multiple nodules, while the evidence for thyroid cancer risk was “strong” in patients with GD and thyroid nodules compared to GD patients without nodules [39]. In addition, we found strong evidence of an increased risk of thyroid cancer mortality in GD patients compared to non-GD patients (including both euthyroid and hyperthyroid patients) that increased after excluding low-rate incidental cancers and even after stratification by continent (Europe higher than Asia) [41].

An association between GD and thyroid cancer was first hypothesized in the literature in 1955 [46]. Since then, the question of the presence of this association and its effect on prognosis has been a matter of numerous disagreements. Over time, several researchers have sought to understand whether this association was causal, and several possible explanatory mechanisms have been suggested.

GD is known to result from an immune disorder in which the patient’s immunoglobulins bind to thyroid-stimulating hormone receptors, causing hyperthyroidism, promoting tumor formation, causing angiogenesis [40,47], upregulating various growth factors, and enhancing tumor invasiveness [12,48]. Furthermore, autoimmunity of GD and altered host immune tolerance further increase the risk of thyroid cancer [4,10].

Fork head box P3 (Foxp3) gene expression has been observed to be associated with the development of autoimmune diseases and has a suppressive role in the organized immune response against cancer cells [49]. Some authors have shown an association between increased expression of Foxp3 and increased tumor aggressiveness in patients with thyroid cancer [50,51]. However, its levels in thyroid autoimmunity are unknown [52]. Recently, efforts have been made to unravel possible molecular markers associated with the progression of thyroid cancers. Expression of the programmed deathligand-1 (PD-L1) gene in tumor cells has been shown to be associated with increased aggressiveness of tumor characteristics and poor prognosis in patients with thyroid cancer [53]. The expression of this marker has also been demonstrated in patients with GD, and the possibility that this may favor the onset of thyroid cancer cannot be excluded [54]. Other molecular markers may explain the association between inflammation and tumorigenesis. The presence of interleukin-4 (IL-4) in tumor cells is known to modulate cell survival, proliferation, and metastasis [55,56]. Importantly, IL-4 has also been identified in GD [57], thus pointing out another potential mechanism linking the two diseases. MicroRNAs (miRNAs) are non-coding RNAs that regulate gene expression and are known to be involved in diverse cell functions, such as proliferation, apoptosis, differentiation, and tumorigenesis [46]. Analysis of miRNA expression in GD patients showed an intermediate level of expression of miRNAs (miR-146b, miR-221, and miR-222) also expressed in DTC. Thus, a role for these miRNAs in the development of DTC in patients with GD has been suggested [46]. Overall, these findings raise the intriguing hypothesis that molecular dysregulation occurs in GD patients and, at the same time, predisposes these patients to cancer development. Therefore, the existence of a causal link between the two diseases cannot be excluded. However, more evidence is needed to clarify the nature of the association between GD and thyroid cancer.

### 4.2. Strengths and Limitations

This is the first umbrella review that systematically explores the association between GD and thyroid cancer risk and prognosis. Screening and selection of the article outcomes were carried out by four authors. Studies were evaluated using AMSTAR-2.

However, some limitations must be acknowledged in this umbrella review. First, only English-language articles were included in this review and the search has been limited to publications from the past 10 years. Nevertheless, this should not be considered a source of bias because of the adoption of recommended practices or diagnoses changes in time. Secondly, our results come from meta-analyses, which do not include randomized controlled studies, but only observational studies, which have a higher potential for bias and confounding issues [39,40]. Thirdly, heterogeneity is present in the included studies and only some authors have hypothesized the reasons for this [38,39,40,41,42]. Fourth, the comparison groups are not homogeneous across the included studies. Fifth, the follow-up consists of different periods, so it is difficult to hypothesize how it is really involved in this process. Sixth, there is no consensus in defining the study design, especially for prospective and retrospective studies. Fifth, many studies did not report the type of partial or total thyroidectomy surgery type [39,40] or data by the gender of participants [39,40,41,42]. Finally, it was impossible to estimate the prevalence of medullary thyroid cancer (MTC) in GD patients. While papillary carcinomas are the most common histological types of thyroid cancer, followed by follicular carcinomas, MTC is less frequent. Only 15 cases of MTC have been reported in patients with GD so far [58].

## 5. Conclusions

In conclusion, this umbrella review found “strong” evidence for the risk of thyroid cancer in patients having both GD and thyroid nodules and for the risk of thyroid cancer mortality in patients with GD, particularly in Europe. The results of this umbrella review should be interpreted with caution; as the evidence comes mainly from retrospective studies, potential concerns are selection and recall bias, and whether the empirically observed association reflects a causal relationship remains an open question. Finally, even if our review did not find definitive results, it allowed us to highlight the limitations of the current studies, and this will be useful for carrying out future studies of better quality.

## Figures and Tables

**Figure 1 cancers-15-02724-f001:**
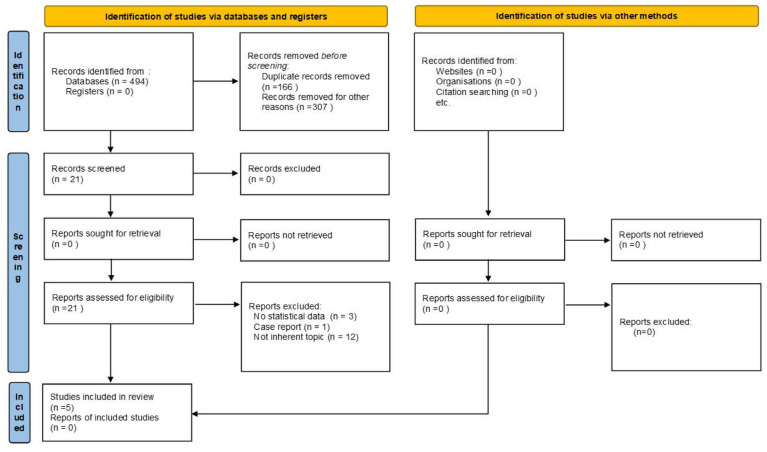
Flow chart of identification and selection studies.

**Table 1 cancers-15-02724-t001:** Criteria used to evaluate the strength of evidence.

Evidence	Criteria Used
Strong	OR * > 2; *p* ** < 10^−6^; >1000 cases; *p* < 0.05 of largest study in meta-analysis; I^2^ *** < 50%; no small study effect; prediction interval excludes null value **; no excess significance bias
Moderate	OR > 1.5; *p* * < 10^−6^; >1000 cases; *p* < 0.05 of largest study in meta-analysis
Modest	OR >1.2; *p* * < 10^−3^
Weak	OR > 1; *p* < 0.05

* OR: oddsratio. ** As suggested by the American Statistical Society (ASA); in the case of a lower confidence interval limit with a value greater than 0.70, we have considered the effect found as a trend to watch [35,36].*** I^2^ index; it describes the percentage of variation across studies that is due to heterogeneity rather than chance.

**Table 2 cancers-15-02724-t002:** Quality assessment and risk of bias in systematic reviews using “A Measurement Tool to Assess Systematic Reviews” (AMSTAR 2).

	Mekraksakit et al., 2019 [38]	Papanastasiou et al., 2019 [39]	Staniforth et al., 2015 [40]	Song et al., 2019 [41]	Jia et al., 2018 [42]
Did the research questions and inclusion criteria for the review include the components of PICO?	Yes	Yes	Yes	Yes	Yes
2. Did the report of the review contain an explicit statement that the review methods were established prior to the conduct of the review and did the report justify any significant deviations from the protocol? *	Partial Yes	Partial Yes	Partial yes	Partial Yes	Partial Yes
3. Did the review authors explain their selection of the study designs for inclusion in the review?	Yes	No	Yes	Yes	Yes
4. Did the review authors use a comprehensive literature search strategy? *	Partial Yes	Partial Yes	Partial Yes	Partial Yes	Partial Yes
5. Did the review authors perform study selection in duplicate?	Yes	Yes	No	Yes	Yes
6. Did the review authors perform data extraction in duplicate?	Yes	Yes	No	Yes	Yes
7. Did the review authors provide a list of excluded studies and justify the exclusions?	No	No	No	No	No
8. Did the review authors describe the included studies in adequate detail? *	Yes	Partial Yes	No	Yes	Partial Yes
9. Did the review authors use a satisfactory technique for assessing the risk of bias (RoB) in individual studies that were included in the review? *	Yes	Yes	Yes	Partial Yes	Partial Yes
10. Did the review authors report on the sources of funding for the studies included in the review?	Yes	Yes	Yes	Yes	Yes
11. If meta-analysis was performed, did the review authors use appropriate methods for statistical combination of results? *	Yes	Yes	Yes	Yes	Yes
12. If meta-analysis was performed, did the review authors assess the potential impact of RoB in individual studies on the results of the meta-analysis or other evidence synthesis?	Yes	Yes	Yes	Yes	Yes
13. Did the review authors account for RoB in primary studies when interpreting/discussing the results of the review? *	Yes	Yes	Yes	Yes	Yes
14. Did the review authors provide a satisfactory explanation for, and discussion of, any heterogeneity observed in the results of the review?	Yes	Yes	Yes	Yes	No
15. If they performed quantitative synthesis, did the review authors carry out an adequate investigation of publication bias (small study bias) and discuss its likely impact on the results of the review? *	Yes	Yes	Yes	No	Yes
16. Did the review authors report any potential sources of conflict of interest, including any funding they received for conducting the review?	Yes	Yes	Yes	Yes	Yes
Total of yes	13/16 (75.0%)	11/16 (68.7%)	10/16 (75.0%)	11/16 (68.7%)	10/16 (62.5%)
Rating overall confidence	Moderate	Moderate	Low	Low	Moderate

* Critical AMSTAR domain.

**Table 3 cancers-15-02724-t003:** Summary of evidence for meta-analyses on the risk and prognosis of thyroid cancer in patients with Graves’ disease.

Author, Year	N. and Study Design in Meta-Analyses	Graves’ Patients (GD) (Number of Thyroid Cancer Cases)	Sex by Comparison Groups	Comparison Groups (N)	Age (Years)	Type of Thyroid Cancer	Country	Period of Primary Studies’ Publication	Exposure Time Period/ Follow-Up	Effect Size by Comparing Group (Thyroid Cancer Risk in GD Patients, Mortality, Recurrence/Persistence)	Heterogenicity (*p*-Value) *	Publication Bias
Mekraksakit et al., 2019 [38]	15 retrospective cohorts 9 case-control 1 prospective cohort	GD: 2892(2892)	F: 1320 M: 1572	-DTC patients with non-Graves’ hyperthyroidism -DTC patients with euthyroidism -Non-specified DTC patients -DTC with TNMG -DTC with TA	from 5 to 81	2662 PTC, 213 FTC, 16 mixed PTC and FTC 1 CCC	1 Germany, 4 USA, 7 Italy, 2 Taiwan, 2 Turkey, 1 Oman, 1 Greece, 1 U.K., 1 Spain, 1 Australia, 2 Japan, 1 India	from 1988 to 2018	from 1 to 30 years	**GD vs. no GD** hyperthyroidism		Yes
										OR for mortality = 0.79 (95% CI 0.17–3.67)	0.0% (0.50)	
										OR for recurrence/persistence 2.66 (95% CI 0.94–7.54)	1.8% (0.41)	
										**GD vs. euthyroid**		
										OR for mortality = 2.69 (95%CI 0.70–10.40)	22.9% (0.27)	
										OR for recurrence/persistence = 1.39 (95% CI 0.52–3.76)	64.2% (0.04)	
										**GD vs. Non-specified DTC** OR for recurrence/persistence = 0.91 (95% CI 0.18–4.58)	73.3% (0.01)	
Papanastasiou et al., 2019 [39]	7 retrospectivecohorts	GD:2582(297)	F: 1368 M: 517 (this number is partly due to the lack of data in some of the included studies)	-GD patients without thyroid nodules -GD patients without thyroid nodules (without malignant or suspicious cytology) -GD patients with multiple nodules	from 27 to 58	297 DTC	1 France, 4 Turkey, 1 USA, 1 China	from 1988 to 2018	Notapplicable	**GD with thyroid nodules vs. GD without thyroid nodules** OR incidence for thyroid cancer risk = 5.30 (95% CI 2.43–11.59)	83%(0.00)	Not evaluated because of the insufficient number of included studies
										**GD with thyroid nodules vs. GD without malignant or suspicious cytology** OR incidence for thyroid cancer risk = 4.02 (95% CI 1.24–12.99)	89% (0.00)	
										**GD with thyroid nodules vs. number of nodules** OR incidence for thyroid cancer risk = 1.39 (95% CI 0.85–2.29)	0% (0.76)	
Staniforth et al., 2015 [40]	28 retrospective 1 cohort 3 prospective 1 case-control	GD: 10,594 (498)	GD: 451 M, 2456 F UTG: 674 M, 791 F MTG: 276 M, 491 F Hyperthyroidism/Thyrotoxicosis: 199 M, 573 F Goiter: 1613 M, 12,887 F (this number is partly due to the lack of data in some of the included studies)	Patients with non-Graves’ hyperthyroidism: -Any type of toxic nodular goiter -Toxic multinodulargoiter-Toxic uninodulargoiter-Unspecified toxic nodular goiter	from 3 to 82	**325 out of 498 cases had the histological diagnosis**: Papillary: 286 (88%) Follicular: 34 (10%) Mixed papillary-follicular: 2 (0.6%) Medullary: 2 (0.6%) Anaplastic: 1 (0.3%)	7 Asia, 18 Europe, 2 Pacific Area, 6 USA	1977–2014	from 2 to 25 years	**GD vs. any type of toxic nodular goiter** OR incidence = 0.89 (95% CI 0.63–1.26)	28.57% (0.10)	No publication bias (*p* = 0.98)
										**GD vs. toxic multinodular goiter** OR incidence= 1.24 (95% CI 0.81–1.90)	0.0% (0.82)	
										**GD vs. toxic unimodular goiter** OR incidence = 0.96 (95% CI 0.58–1.57)	5.13% (0.39)	
										**GD vs. unspecified toxic nodular goiter** OR incidence = 0.43 (95% CI 0.14–1.33)	71.73% (0.01)	
Song et al., 2019 [41]	12 retrospective	GD: 882 (36)	GD: 189 M, 1345 F UTG: 23 M, 141 F MTG: 166 M, 929 F Hyperthyroidism: 15 M, 68 F DTC: 22 M, 117 FEuthyroidism: 28 M, 181 F Thyroidectomy not GD and PTC: 104 M, 405 F Thyroidectomy not GD and STC: 33 M, 476 F (this number is partly due to the lack of data in some of the included studies)	-Non-Graves’ DTC patients (N: 2201) -Non-Graves’ hyperthyroidism DTC patients (N: 118) -Euthyroidism DTC patient (N: 697)	from 15 to 51 (meanage)	2708 PTC 31 FTC 159 DTC	3 Italy, 2 Greece, 2 USA, 1 India, 2 Japan, 1 U.K., 1 China	from 1988 to 2018	Notapplicable	**GD patients vs. not GD patients** OR(recurrence/disease progress/persistence) = 1.07 (0.51–2.22)	65% (0.00)	Not reported
										**Moderate–high quality subgroup** OR(recurrence/disease progress/persistence) = 1.50 (95% Cl 0.60–3.79)	64% (0.00)	
										**Weak quality subgroup** OR (recurrence/disease progress/persistence) = 0.53 (0.20–1.43)	51% (0.13)	
										**By K-M curves** OR(recurrence/disease progress/persistence) = 2.02 (95% Cl 1.04–3.90)	0% (0.04)	
										**Europe** OR (recurrence/disease progress/persistence) = 1.77 (95% Cl 0.99–3.16)	0% (0.47)	
										**Europe and America** OR (recurrence/disease progress/persistence) = 1.74 (95% Cl 1.02–2.98)	N.A.	
										**Asia** OR (recurrence/disease progress/persistence) = 0.43 (95% Cl 0.25–0.77)	80% (0.00)	
										**Retrospective not randomized studies with subgroup** OR(recurrence/disease progress/persistence) = 0.50 (95% Cl 0.30–0.85)	76% (0.00)	
										**Retrospective randomized studies subgroup** OR(recurrence/disease progress/persistence) = 1.79 (95% Cl 1.01–3.18)	0% (0.47)	
										**High incidental carcinoma rate studies** OR(recurrence/disease progress/persistence) = 1.75 (95% Cl 1.04–2.95)	0% (0.81)	
										**GD vs. not GD hyperthyroidism** OR(recurrence/disease progress/persistence) = 3.56 (95% Cl 1.18–10.75)	5% (0.37)	
										**GD vs. euthyroid** OR(recurrence/disease progress/persistence) = 0.86 (95% Cl 0.42–1.77)	93% (0.00)	
										**GD vs. not GD** OR for mortality = 2.93 (95% Cl 1.17–7.37)	33% (0.20)	
										**High incidental carcinoma rate studies** OR for mortality = 7.17 (95% Cl 2.14–24.02)	0% (0.51)	
										**Europe** OR for mortality = 4.89 (95% Cl 1.52–15.75)	38% (0.20)	
										**Asia** OR for mortality = 1.13 (95% Cl 0.21–6.13)	0% (0.34)	
										**Retrospective not randomized studies with subgroup** OR for mortality = 3.75 (95% Cl 1.29–10.90)	57% (0.10)	
										**Retrospective randomized studies subgroup** OR for mortality = 1.36 (95% Cl 0.19–9.82)	0% (0.43)	
										**GD vs. euthyroid** OR for mortality = 3.99 (95% Cl 1.19–13.39)	78% (0.03)	
										**GD vs. not GD hyperthyroidism** OR for mortality= 1.36 (95% Cl 0.19–9.82)	0% (0.43)	
Jia et al., 2018 [42]	11 cohorts	10743 (207)	GD patients with PTC GD: 1065 F, 9678 M (this number is partly due to the lack of data in some of the included studies)	TA patients with TC TNG patients with TC non-GD patients with PTC	from 17 to 76	207 DTC	2 USA, 1 Oman, 1 Greece, 1 Turkey, 1 India, 3 Italy, 1 Germany, 1 France	from 1946 to 2013	Not reported	**Surgery-hyperthyroid incidental thyroid cancer patients with GD vs. not GD** OR incidence: 1.0 (0.68–1.46 *p* = 0.98).	12% (0.33)	No publication bias (*p* = 0.77)
										**GD vs. toxic** **adenoma patients** OR incidence: 0.53 (0.21–1.36 *p* = 0.18)	40% (0.17)	
										**GD vs. TNG patients** OR incidence: 1.01 (0.65–1.57 *p* = 0.95)	5% (0.39)	
										**GD patients and non-GD patients** OR incidence: 0.79 (0.24–2.64 *p* = 0.70)	0% (0.97)	
										**GD vs. Toxic multinodular goiter** OR incidence = 1.24 (95% CI 0.81–1.90)	0.0% (0.82)	
										**GD vs. Toxic unimodular goiter** OR incidence = 0.96 (95% CI 0.58–1.57)	5.13% (0.39)	
										**GD vs. unspecified toxic nodular goiter** OR incidence = 0.43 (95% CI 0.14–1.33)	71.73% (0.01)	

* I^2^ test interpretation:from 0 to 40%, might not be important; from 30 to 60%, may be moderate; from 50 to 90%, may be substantial; from 75 to 100%, may be considerable. In general, the test is conservative and so a non-significant result cannot be interpreted as showing that there is no heterogeneity. For this reason, the cutoff *p* < 0.10 was used rather than *p* < 0.05 to indicate heterogeneity [43]. Note: MTG: multinodular toxic goiter, UTG: uninodular toxic goiter, TNG: toxic nodular goiter, uTNG: unspecified toxic nodular goiter, TA: toxic adenoma, M: males, F: females, CCC: clear cell carcinoma.

## Data Availability

Not applicable.

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
