# Peer review of "Risk and Prognosis of Thyroid Cancer in Patients with Graves’ Disease: An Umbrella Review"

_cancers, 2023, doi:10.3390/cancers15102724_

Round 1

Reviewer 1 Report

Within an umbrella review the authors explored the evidence supporting the association between Graves’ Disease (GD) and the risk of developing thyroid cancer risk and its prognosis. They found strong evidence for thyroid cancer risk in GD patients and nodular thyroid disease and thyroid cancer mortality risk in GD patients, particularly in Europe. Overall, the review is interesting and well written, and addressed the purpose with clearly presented and discussed results. I only have minor comments.

A section describing statistical analysis and significance as well as I2 test interpretation should be added in the Methods.

Lines 170-172 and lines 198-200. Please clarify this point.

Page 11, Discussion, lines 14-15. I think that “and” should be replaced with “compared with”.

Page 12, line 55. Please change to “to be associated”.

Page 12, line 86. Defying or defining?

Page 12, line 94. Replace with “of which”.

Author Response

Reviewer 1

Within an umbrella review the authors explored the evidence supporting the association between Graves’ Disease (GD) and the risk of developing thyroid cancer risk and its prognosis. They found strong evidence for thyroid cancer risk in GD patients and nodular thyroid disease and thyroid cancer mortality risk in GD patients, particularly in Europe. Overall, the review is interesting and well written, and addressed the purpose with clearly presented and discussed results. I only have minor comments.

A section describing statistical analysis and significance as well as I2 test interpretation should be added in the Methods.

The Reviewer makes a valid point; therefore, we have added a section describing statistical analysis and significance as well as I2 test interpretation in the Methods.

Lines 170-172 and lines 198-200. Please clarify this point.

The Reviewer makes a valid point; therefore, we have rephrased both sentences

Comments on the Quality of English Language

Page 11, Discussion, lines 14-15. I think that “and” should be replaced with “compared with”.

Done

Page 12, line 55. Please change to “to be associated”.

Done

Page 12, line 86. Defying or defining?

We have replaced defying with defining

Page 12, line 94. Replace with “of which”.

Done

Reviewer 2 Report

The study by Palella and collaborators analyses the potential connection between thyroid cancer and Graves’ disease. This issue is generally interesting and has been studied for many years. I am not sure if the conclusions of this paper add too much to the field. Minor comments are listed below.

1.       It is worth mentioning medullary thyroid cancer (MTC). Please briefly explain why there are few studies examining the relationship between Graves’ disease and MTC.

2.       Please explains all acronyms at the first mention, e.g., OR and I2 in Table 1, TNG, TA, M, F and CCC and in Table 3.

3.       Line 66: I would suggest using full names of miRNA, e.g. “miRNA 146b” or “miR-146b” in place of “146b”.

Examples of typos:

1.       Please remove unnecessary spaces, e.g., (i) “0 %” in lines 207, 208 and in Table 3, (ii) “33 %” in line 223, (iii) “p= 0.41” in line 236, etc.

2.       Please add missing spaces, e.g., “havebeenmade” in line 53, “theprogrammed” in line 54, “Mekraksakitet al., 2019” and “Papanastasiouet al., 2019[39]” in Table 2, etc.

3.       The Authors use versus as ‘vs.’ (with a dot) and ‘vs’ (without a dot). Please unify it.

4.       Lines 214 and 216: Change “p 0.27” and “p 0.03” for “p=0.27” and “p=0.03”, respectively. Similar change is needed in lines 207-208.

5. Lines 190, 191, etc: I am not sure if expression "p=0.00" is correct. Maybe it's better to use "p<0.01"?

The study by Palella and collaborators analyses the potential connection between thyroid cancer and Graves’ disease. This issue is generally interesting and has been studied for many years. I am not sure if the conclusions of this paper add too much to the field. Minor comments are listed below.

1.       It is worth mentioning medullary thyroid cancer (MTC). Please briefly explain why there are few studies examining the relationship between Graves’ disease and MTC.

2.       Please explains all acronyms at the first mention, e.g., OR and I2 in Table 1, TNG, TA, M, F and CCC and in Table 3.

3.       Line 66: I would suggest using full names of miRNA, e.g. “miRNA 146b” or “miR-146b” in place of “146b”.

Examples of typos:

1.       Please remove unnecessary spaces, e.g., (i) “0 %” in lines 207, 208 and in Table 3, (ii) “33 %” in line 223, (iii) “p= 0.41” in line 236, etc.

2.       Please add missing spaces, e.g., “havebeenmade” in line 53, “theprogrammed” in line 54, “Mekraksakitet al., 2019” and “Papanastasiouet al., 2019[39]” in Table 2, etc.

3.       The Authors use versus as ‘vs.’ (with a dot) and ‘vs’ (without a dot). Please unify it.

4.       Lines 214 and 216: Change “p 0.27” and “p 0.03” for “p=0.27” and “p=0.03”, respectively. Similar change is needed in lines 207-208.

5. Lines 190, 191, etc: I am not sure if expression "p=0.00" is correct. Maybe it's better to use "p<0.01"? 

Author Response

Reviewer 2

The study by Palella and collaborators analyses the potential connection between thyroid cancer and Graves’ disease. This issue is generally interesting and has been studied for many years. I am not sure if the conclusions of this paper add too much to the field.

The Reviewer makes a valid point; therefore, we have specified in the conclusions that even if our review did not find definitive results, it allowed us to highlight the limitations of the current studies, this will be useful for carrying out future studies of better quality.

Minor comments are listed below.

  1. 1.       It is worth mentioning medullary thyroid cancer (MTC). Please briefly explain why there are few studies examining the relationship between Graves’ disease and MTC.

      The Reviewer makes a valid point; therefore, we included the explanation in the last section of the strengths and limitations paragraph.

  1. Please explains all acronyms at the first mention, e.g., OR and I2 in Table 1, TNG, TA, M, F and CCC and in Table 3.

Done

  1. Line 66: I would suggest using full names of miRNA, e.g. “miRNA 146b” or “miR-146b” in place of “146b”. 

 Done

Examples of typos:

  1. Please remove unnecessary spaces, e.g., (i) “0 %” in lines 207, 208 and in Table 3, (ii) “33 %” in line 223, (iii) “p= 0.41” in line 236, etc.

Done

  1. Please add missing spaces, e.g., “havebeenmade” in line 53, “theprogrammed” in line 54, “Mekraksakitet al., 2019” and “Papanastasiouet al., 2019[39]” in Table 2, etc.

Done

  1. The Authors use versus as ‘vs.’ (with a dot) and ‘vs’ (without a dot). Please unify it.

Done

  1. Lines 214 and 216: Change “p 0.27” and “p 0.03” for “p=0.27” and “p=0.03”, respectively. Similar change is needed in lines 207-208.

Done

  1. Lines 190, 191, etc: I am not sure if expression "p=0.00" is correct. Maybe it's better to use "p<0.01"?

We have chosen to use the real p-value calculated by the Statistical Package for Social Science (SPSS) rather than the threshold p<0.01 because we believe it is more correct.

Reviewer 3 Report

This paper entitled “Risk and prognosis of thyroid cancer in patients with Graves’ disease: an umbrella review” by Dr. Marco Palella et al. decribes a review to assess the quality and relevance of the evidence on the association between Graves' disease and thyroid cancer risk and prognosis, including mortality, recurrence and persistence rates.

1.    In Table 2, if "Partial yes" is not counted as "yes," then "In total of yes" should be 13/16 instead of 12/16 in the column of Mekraksakite et al. Furthermore, "In total of yes" in the column of Staniforth et al. is 12/16, but it should be 10/16.

2.    For example, there is no spacing in the following parts. Please review through the full text.

There is no space before et al. in Table 2.

In Discussion part, line 53, havebeenmade.

The sama part, line 54, theprogrammed.

The same part, line 65, anintermediate.

3.    The 2 in I2 should be superscripted.

This point is described in Comments to the Authors.

Author Response

Reviewer 3

This paper entitled “Risk and prognosis of thyroid cancer in patients with Graves’ disease: an umbrella review” by Dr. Marco Palella et al. decribes a review to assess the quality and relevance of the evidence on the association between Graves' disease and thyroid cancer risk and prognosis, including mortality, recurrence, and persistence rates.

  1. In Table 2, if "Partial yes" is not counted as "yes," then "In total of yes" should be 13/16 instead of 12/16 in the column of Mekraksakite et al. Furthermore, "In total of yes" in the column of Staniforth et al. is 12/16, but it should be 10/16.

      The Reviewer makes a valid point; therefore, we have corrected the errors pointed out by the reviewer.

  1. For example, there is no spacing in the following parts. Please review through the full text.

There is no space before et al. in Table 2.

In Discussion part, line 53, havebeenmade.

The sama part, line 54, theprogrammed.

The same part, line 65, anintermediate.

The Reviewer makes a valid point; therefore, we have reviewed the full text.

  1. The 2 in I2 should be superscripted.

Done.
